# A screening of the MMV Pathogen Box® reveals new potential antifungal drugs against the etiologic agents of chromoblastomycosis

**Rowena Alves Coelho**[1]*, **Luna Sobrino Joffe**[2], **Gabriela Machado Alves**[1], **Maria Helena Galdino Figueiredo-Carvalho**[1], **Fábio Brito-Santos**[1], **Ana Claudia Fernandes Amaral**[3], **Marcio L. Rodrigues**[4,5], **Rodrigo Almeida-Paes**[1]

**1** Mycology Laboratory at the Evandro Chagas National Institute of Infectious Diseases, Oswaldo Cruz Foundation (Fiocruz), Rio de Janeiro, Brazil, **2** Department of Microbiology and Immunology, Stony Brook University, New York, NY, United States of America, **3** Laboratory of Natural Products Chemistry, Farmanguinhos, Fiocruz, Rio de Janeiro, Brazil, **4** Carlos Chagas Institute, Fiocruz, Paraná, Brazil, **5** Microbiology Institute, Federal University of Rio de Janeiro (UFRJ), Rio de Janeiro, Brazil

* rowena.alves@ini.fiocruz.br, rodrigo.paes@ini.fiocruz.br

## Abstract

Chromoblastomycosis (CBM) is a chronic subcutaneous mycosis caused by traumatic implantation of many species of black fungi. Due to the refractoriness of some cases and common recurrence of CBM, a more effective and less time-consuming treatment is mandatory. The aim of this study was to identify compounds with *in vitro* antifungal activity in the Pathogen Box® compound collection against different CBM agents. Synergism of these compounds with drugs currently used to treat CBM was also assessed. An initial screening of the drugs present in this collection at 1 µM was performed with a *Fonsecaea pedrosoi* clinical strain according to the EUCAST protocol. The compounds with activity against this fungus were also tested against other seven etiologic agents of CBM (*Cladophialophora carrionii*, *Phialophora verrucosa*, *Exophiala jeanselmei*, *Exophiala dermatitidis*, *Fonsecaea monophora*, *Fonsecaea nubica*, and *Rhinocladiella similis*) at concentrations ranging from 0.039 to 10 µM. The analysis of potential synergism of these compounds with itraconazole and terbinafine was performed by the checkerboard method. Eight compounds inhibited more than 60% of the *F. pedrosoi* growth: difenoconazole, bitertanol, iodoquinol, azoxystrobin, MMV688179, MMV021013, trifloxystrobin, and auranofin. Iodoquinol produced the lowest MIC values (1.25–2.5 µM) and MMV688179 showed MICs that were higher than all compounds tested (5 - >10 µM). When auranofin and itraconazole were tested in combination, a synergistic interaction (FICI = 0.37) was observed against the *C. carrionii* isolate. Toxicity analysis revealed that MMV021013 showed high selectivity indices (SI ≥ 10) against the fungi tested. In summary, auranofin, iodoquinol, and MMV021013 were identified as promising compounds to be tested in CBM models of infection.

**Data Availability Statement:** All relevant data are within the manuscript and its Supporting Information files.

**Funding:** This work has received major financial support from Inova Fiocruz/VPPCB (VPPCB-008-FIO-18-2-49). MLR acknowledges support from the Brazilian agency Conselho Nacional de Desenvolvimento Científico e Tecnológico (CNPq, grants 405520/2018-2, 440015/2018-9, and 301304/2017-3) and Inova Fiocruz/VPPCB and Inova Fiocruz/VPPIS (grants VPPCB-007-FIO-18-2-57 and VPPIS-001-FIO-18-66). He also acknowledges support from Coordenação de Aperfeiçoamento de Pessoal de Nível Superior (CAPES, finance code 001) and the Instituto Nacional de Ciência e Tecnologia de Inovação em Doenças de Populações Negligenciadas (INCT-IDPN). The funders had no role in study design, data collection and analysis, decision to publish, or preparation of the manuscript.

**Competing interests:** The authors have declared that no competing interests exist.

# Introduction

Chromoblastomycosis (CBM) is a subcutaneous neglected mycosis [1]. It is caused by the implantation of one of its etiological agents through a trauma to the skin. Black fungi from the genera *Fonsecaea*, *Cladophialophora*, *Rhinocladiella*, *Phialophora*, *Exophiala*, among others, can cause CBM. Since this mycosis is not of compulsory notification in most countries where it occurs, its distribution is based on case reports and case series, with most of them occurring in Latin America, Africa, and Asia. In South America, most of the cases of CBM have been described in Brazil, in Africa most cases come from Madagascar, and China is the leading Asian country on number of cases [2]. In Brazil, the most frequent CBM agents belong to the genus *Fonsecaea* [3–6]. *F. pedrosoi* is the predominant species in South America, followed by *F. monophora* [7]. *F. pedrosoi* also predominates in humid regions of most countries where CBM is endemic, including Madagascar and China. *Cladophialophora carrionii* is more frequent in arid areas of these countries [8–10]. The frequency of infections by species belonging to other genera of black fungi varies among different geographic regions [2].

CBM mainly affects agricultural and construction workers, which develop extensive injuries with damage to the affected limb [11]. Most patients often take a long time to seek medical help, therefore their lesions are usually extensive [2]. Although there is no official therapeutic protocol for CBM, itraconazole is the most frequently used drug, followed by terbinafine [2]. In developing countries, including Brazil, treatment based on the use of these drugs is long and expensive, and some patients show recurrence and refractoriness [2,12]. These patients usually require more than one therapeutic method, often including physical methods, such as cryosurgery or laser therapy [13]. The *in vitro* response of the CBM agents to other antifungal drugs currently used to treat mycotic infections, such as amphotericin B, fluconazole, flucytosine, and micafungin is not satisfactory [14]. A more effective and less time-consuming treatment is clearly required to combat CBM.

The discovery of new pharmacological agents, however, is costly and time-consuming. Moreover, most of compounds in pre-clinical or clinical studies will never be approved for human use [15]. A useful approach to bypass these problems is drug repurposing, where drugs already studied, and sometimes approved, to treat other medical conditions are redirected to target a new disease [16].

In order to identify novel drugs with activity against neglected diseases, the Medicines for Malaria Venture (MMV, Switzerland; http://www.pathogenbox.org/), developed a collection of 400 compounds named Pathogen Box®. This initiative provides the tools for identifying active compounds against neglected pathogens [17,18]. The aim of this study was to identify *in vitro* antifungal activity within the Pathogen Box® compounds against the CBM agents and to investigate the synergism of these new compounds with drugs currently used to treat CBM.

# Materials and methods

## Fungal strains and growth conditions

The eight strains used in the study were obtained from the Collection of Pathogenic Fungi (CFP) of Fiocruz, as well as from the American Type Culture Collection (ATCC). *Fonsecaea pedrosoi* CFP00791 was used throughout the study. This species is one of the major agents of CBM in most regions where this disease is endemic. *Cladophialophora carrionii* CFP 00910, *Phialophora verrucosa* CFP 00937, *Fonsecaea monophora* CFP 00911, *Fonsecaea nubica* CFP 00912, *Rhinocladiela similis* CFP 00790, *Exophiala jeanselmei* var. *heteromorpha* ATCC 28180, and *Exophiala dermatitidis* ATCC 28869 were used for minimal inhibitory concentration (MIC) and synergism assays. Strains were maintained on potato dextrose agar (PDA) (Sigma

Chemical Corporation, St. Louis, MO, USA). Seven-day-old cultures incubated at 30°C were used in the assays.

## The Pathogen Box® compounds

The Pathogen Box® was kindly provided by Medicines for Malaria Venture (MMV, Geneva, Switzerland). It contains 400 different compounds tested for cytotoxicity with values within levels considered acceptable for an initial drug discovery programme [19]. The Pathogen Box® contains compounds with proven activity against neglected pathogens, including those causing tuberculosis, malaria, helminthiasis, cryptosporidiosis, toxoplasmosis, and dengue [20–23]. The compounds were supplied in 96-well microtiter plates containing 10 μl/well of 10 mM compound solutions in dimethylsulfoxide (DMSO). The plates were diluted to a final drug concentration of 1 mM in DMSO (Sigma Chemical Corporation, St. Louis, MO, USA), as recommended by the fabricant, for the drug screening. The first and last columns in each plate were left as blank for negative and positive controls, respectively. All plates were stored at −20°C until their use in the following experiments.

## Screening for antifungal activity

For the initial screening, all compounds were tested in 96-well plates (Kasvi Ltda, São José dos Pinhais, PR, Brazil) at a final concentration of 1μM in 100 μl of RPMI 1640 medium, with phenol red, with L-glutamine, and without sodium bicarbonate (Sigma Chemical Corporation), buffered with morpholine propanesulfonic acid (MOPS) (Vetec Química Fina Ltda, Rio de Janeiro, RJ, Brazil) at pH 7.0, and supplemented for a final 2% glucose (Neon Comercial Ltda, São Paulo, SP, Brazil) concentration. The antifungal drugs available as reference compounds in the MMV collection were used as controls of fungal growth inhibition. DMSO concentration in all wells, including those used for fungal control growth, at this point, corresponded to 1%. For preparation of the fungal inoculum, the *F. pedrosoi* strain CFP 00791 was grown as described above and then its conidia were suspended in sterile distilled water supplemented with 0.1% Tween 20 (Sigma Chemical Corporation) and vortexed, with the suspension turbidity adjusted to the 0.5 McFarland scale. This suspension was further diluted 1:10 and then 100 μl of the fungal inoculum was added to each well containing the compounds, generating a final working inoculum density of $2–5 \times 10^5$ CFU/ml and a final DMSO concentration of 0.5% in each well, including controls [21]. Plates were incubated at 35°C for 72–96 hours. The optical density at 530 nm (OD530) was recorded using the Epoch microplate reader (Biotek Instruments Inc, Winooski, VT, USA). The percentage of inhibition of fungal growth (%IG) was calculated according to the formula: %IG = (1- (OD1 / OD2)) × 100, where OD1 = fungal optical density in the presence of the drug; OD2 = optical density of the fungal growth control well without any drug. The %IG data of the 400 compounds was plotted using the Prism 8 software (GraphPad Software, San Diego, CA, USA). Compounds that presented %IG values greater than 60% were selected for further assays.

## Minimal inhibitory concentration (MIC) of selected drugs

Assays for MIC determination were performed in sterile polystyrene flat-bottom 96-well microtiter plates using the broth microdilution method according to the EUCAST guidelines [24]. The final concentration of the selected compounds of the Pathogen Box® ranged from 0.015 to 10 μM. All isolates were tested at a final cellular density of 2 to $5 \times 10^5$ CFU/ml. Plates were incubated at 35°C for 72–96 h. All experiments were repeated at least two times. The MIC was defined as the lowest drug concentration (μM) that inhibited 100% of fungal growth.

## Fungicidal activity

The minimal fungicidal concentration (MFC) was determined by transferring an aliquot of 5 µl of each well without fungal growth of the microdilution plates used for the determination of MIC, as described above, on Sabouraud 2% glucose agar (Sigma Chemical Corporation). The MFC was determined as the lowest drug concentration without fungal growth on the Sabouraud agar after five days of incubation at 35˚C. When the MFC:MIC ratio of an agent is 1 or 2, the compound is considered fungicidal against the pathogen. If the ratio is higher than 2, the mode of action is probably fungistatic [25,26].

## Evaluation of compounds applicability in CBM treatment

Cytotoxicity data of the compounds was provided by the MMV. The selectivity index (SI) was calculated as follows: SI = $CC_{50}$ (µM) / MIC (µM), where $CC_{50}$ is the drug concentration that kills 50% of the cells tested. Compounds with selectivity index lower than 1 were not considered for the synergistic assays. The original use of the compounds and their known side-effects were also verified in the literature, to evaluate their safety for human use. Compounds found to have dangerous side-effects were further excluded from the analysis.

## Synergism evaluation

Synergistic activity between the selected compounds and standard antifungal drugs used in the treatment of CBM (itraconazole and terbinafine) was tested by the checkerboard method [27]. Two drugs were loaded into a single 96-well plate, so that in each of the wells there were different concentrations of the compound-antifungal combination. Compound/antifungal dilutions were prepared following the methodology proposed by the EUCAST, starting from a 100-fold concentrated stock compound/antifungal solution according to the MIC determination methodology [24]. Final concentrations of the standard drugs corresponded to 0.06–4 µg/ml (itraconazole) and 0.015–1 µg/ml (terbinafine). Serial two-fold dilutions of the selected Pathogen Box® compounds were performed so that the MIC of that compound was in a central position of the plate (lines 5–7). Fungal inocula and incubation conditions were the same that were described for the previous experiments. The drug interaction was classified according to the fractional inhibitory concentration index (FICI). The FICI was obtained by the formula: FICI (A/MIC (a)) + (B/MIC (b)), where: A = MIC of the drug (a) in combination; MIC (a) = MIC of drug (a) alone; B = MIC of the drug (b) in combination; MIC (b) = MIC of drug (b) alone [28]. The type of interaction between the antifungal agents in combination was classified as the following: synergism if FICI ≤ 0.5; indifference if 0.5 < FICI < 4 and antagonism if FICI ≥ 4 [27,28].

# Results

## Identification of compounds with antifungal activity against CBM agents

The internal reference drug posaconazole inhibited 99% of the *F. pedrosoi* growth, validating the assay. Amphotericin B, which is also present in the drug collection, inhibited *F. pedrosoi* growth by 29%. Among the 400 compounds present in the Pathogen Box®, another eight drugs were found to have antifungal activity against the *F. pedrosoi* CFP00791 strain (Fig 1). These compounds were identified as difenoconazole (%IG = 98%), bitertanol (%IG = 98%), iodoquinol (%IG = 99%), azoxytrobin (%IG = 95%), MMV688179 (%IG = 73%), MMV021013 (%IG = 62%), trifloxystrobin (%IG = 82%), and auranofin (%IG = 82%). The chemical structure of these compounds is depicted in Fig 2. Their original applications, as described by Medicines for Malaria Venture, are as follows: difenoconazole, bitertanol, MMV688179, and

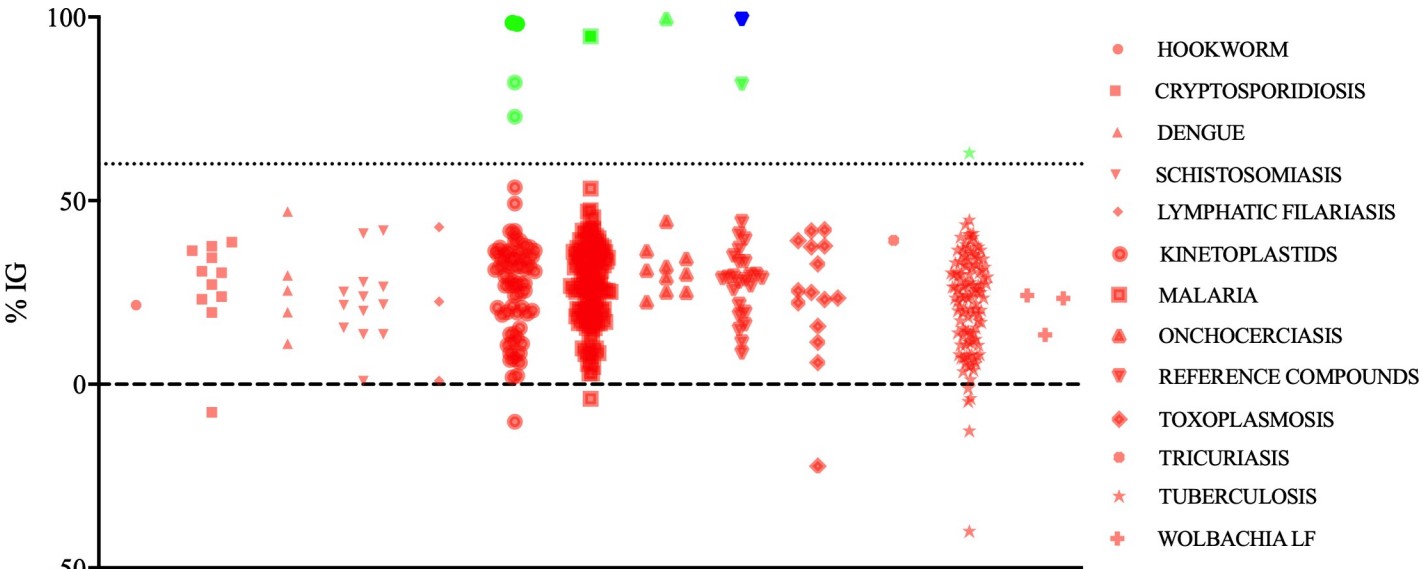

**Fig 1. Screening of 400 compounds present in the MMV Pathogen Box® collection against a *Fonsecaea pedrosoi* strain (CFP00791).** Results are presented as the percent inhibition of growth (%IG) of the fungal strain. The compounds tested are grouped according their original application against some infectious diseases, as described by Medicines for Malaria Venture. The dashed line represents the threshold applied in this study to select compounds with anti-chromoblastomycosis activity. Red symbols represent compounds that were discarded, due to low %IG. Green symbols represent compounds selected for further studies. The blue symbol represents the internal control drug posaconazole.

trifloxystrobin have anti-kinetoplastid action, iodoquinol has anti-onchocerciasis action, azoxystrobin has anti-malaria action, MMV021013 has anti-tuberculosis action, and auranofin, a regulated drug for rheumatoid arthritis treatment, is another reference compound present in the collection.

## Efficacy and spectrum of antifungal activity of the selected compounds

Table 1 depicts the MIC and MFC values for each selected compound and the eight CBM agents tested. In brief, iodoquinol (MMV002817) presented the lowest MIC values among the compounds that we tested for antifungal activity. Seven out of the eight strains had MIC values corresponding to 1.25 µM and one MIC was equal to 2.5 µM to this compound. MMV688179 showed the highest MIC values against all species (5 - >10µM). Auranofin (MMV688978) was fungicidal against *P. verrucosa* and *E. dermatitidis*, and presented fungistatic activity against the other species. Iodoquinol (MMV002817) and bitertanol (MMV688942) were fungicidal against *P. verrucosa* and fungistatic against all other species. MMV688179 was fungicidal against *P. verrucosa* and *F. monophora*, and fungistatic against the other species. All other compounds had fungistatic activity against the set of isolates tested in this study.

## Analysis of synergistic effects

The compound MMV688179 was excluded from further analyses because it showed the highest MIC values (MIC >10 µM) and low selectivity index (Table 2). Trifloxystrobin was found to be toxic to keratinocytes [29], discouraging its use to treat CBM, which affects the skin. From the set of the other agricultural fungicides, we selected bitertanol for synergistic studies, on the basis of its higher selectivity index (Table 2). Therefore, iodoquinol, bitertanol, MMV021013, and auranofin were selected for the determination of FICI in the *in vitro* combination test with itraconazole and terbinafine. The concentration ranges of the compounds

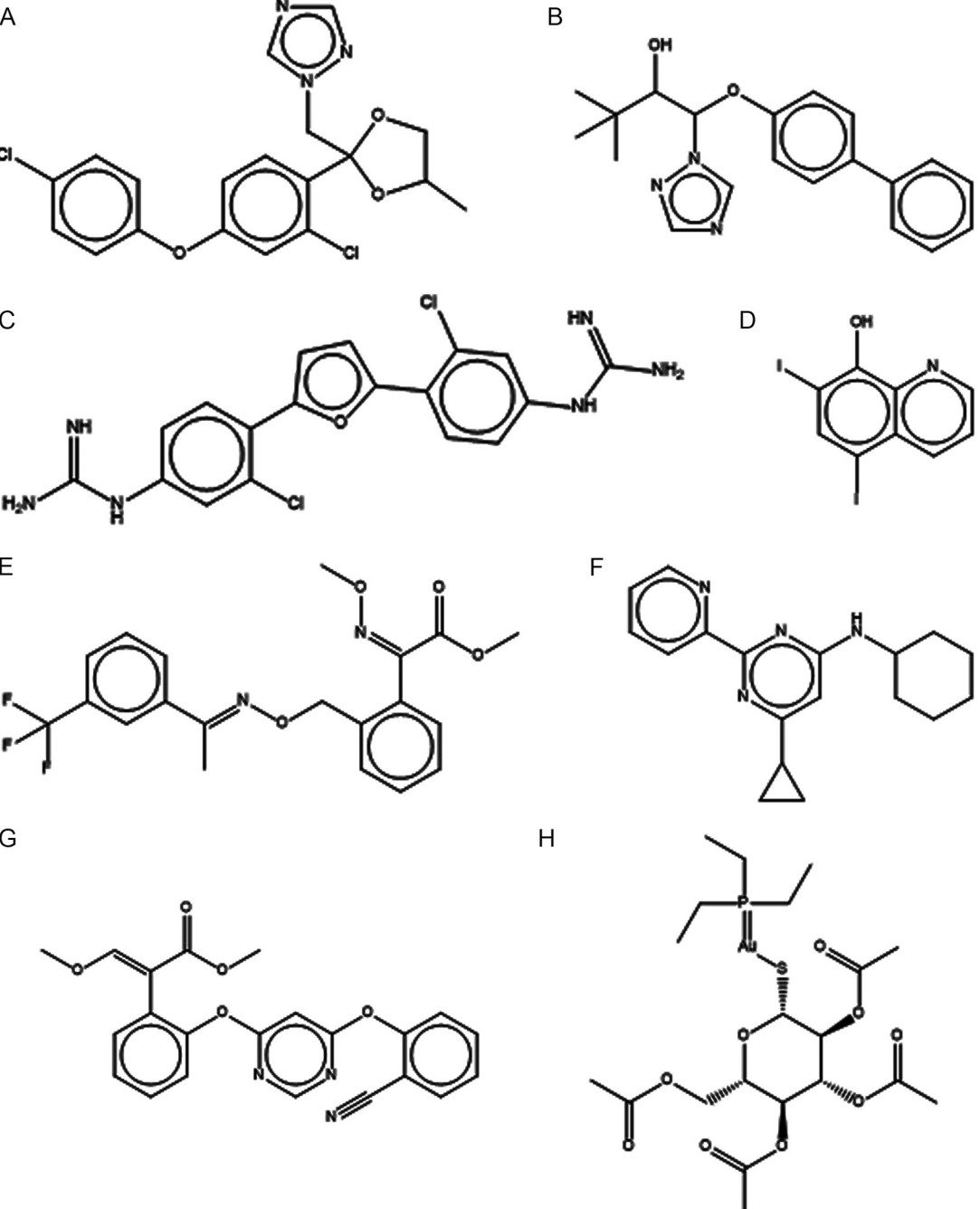

**Fig 2.** Chemical structures of the eight compounds with anti-*Fonsecaea pedrosoi* activity present in the MMV Pathogen Box®
collection: (A) MMV688943 (difenoconazole); (B) MMV688942 (bitertanol); (C) MMV688179 (2-[3-chloro-4-[5-[2-chloro-4-
(diaminomethylideneamino)phenyl]furan-2-yl]phenyl]guanidine); (D) MMV002817 (iodoquinol); (E) MMV688754
(trifloxystrobin); (F) MMV021013 (N-cyclohexyl-6-cyclopropyl-2-pyridin-2-ylpyrimidin-4-amine); (G) MMV021057
(azoxystrobin); (H) MMV688978 (auranofin).

tested corresponded to 0.01–6 µg/ml for auranofin, 0.003–2.5 µg/ml for MMV021013, 0.005–
3 µg/ml for bitertanol, and 0.007–4 µg/ml for iodoquinol. The results obtained from the check-
erboard method with these four compounds in combination with itraconazole or terbinafine

**Table 1. Determination of minimal inhibitory concentration (MIC) and minimal fungicidal concentration (MFC) (μM) against chromoblastomycosis agents.**

| Species | Trifloxystrobin (MMV688754) | | | Auranofin (MMV688978) | | | MMV021013 | | | Azoxystrobin (MMV021057) | | | MMV688179 | | | Difenoconazole (MMV688943) | | | Iodoquinol (MMV002817) | | | Bitertanol (MMV688942) | | |
|---|---|---|---|---|---|---|---|---|---|---|---|---|---|---|---|---|---|---|---|---|---|---|---|---|
| | MIC | MFC | MFC:MIC Ratio | MIC | MFC | MFC:MIC Ratio | MIC | MFC | MFC:MIC Ratio | MIC | MFC | MFC:MIC Ratio | MIC | MFC | MFC:MIC Ratio | MIC | MFC | MFC:MIC Ratio | MIC | MFC | MFC:MIC Ratio | MIC | MFC | MFC:MIC Ratio |
| C. carrionii | 2.5 | >10 | ≥8 | 1.25 | >10 | ≥16 | 5 | >10 | ≥4 | 2.5 | >10 | ≥8 | 5 | >10 | ≥4 | 0.625 | >10 | ≥32 | 1.25 | >10 | ≥16 | 1.25 | 10 | 8 |
| P. verrucosa | 2.5 | >10 | ≥8 | 1.25 | 2.5 | 2 | 5 | >10 | ≥4 | 0.3125 | >10 | ≥64 | 5 | 10 | 2 | 2.5 | 10 | 4 | 1.25 | 1.25 | 1 | 5 | 5 | 1 |
| E. dermatitidis | 10 | >10 | ≥2 | 2.5 | 2.5 | 1 | 5 | >10 | ≥4 | 10 | >10 | ≥2 | >10 | >10 | ≥1 | 2.5 | 10 | 4 | 1.25 | >10 | ≥16 | 10 | >10 | ≥2 |
| E. jeanselmei | 0.625 | >10 | ≥32 | 1.25 | 10 | 8 | 5 | >10 | ≥4 | 0.3125 | >10 | ≥64 | >10 | >10 | ≥1 | >10 | >10 | ≥1 | 1.25 | >10 | ≥16 | 5 | >10 | ≥4 |
| F. pedrosoi | 2.5 | >10 | ≥8 | 1.25 | 10 | 8 | 2.5 | >10 | ≥8 | 1.25 | 10 | 8 | 5 | >10 | ≥4 | 1.25 | 5 | 4 | 1.25 | >10 | ≥16 | 0.625 | 10 | 16 |
| F. monophora | 2.5 | >10 | ≥8 | 1.25 | 5 | 4 | 2.5 | >10 | ≥8 | 2.5 | >10 | ≥8 | 5 | 10 | 2 | 2.5 | 10 | 4 | 1.25 | 10 | 8 | 2.5 | >10 | ≥8 |
| F. nubica | 0.625 | >10 | ≥32 | 1.25 | >10 | ≥16 | 1.25 | >10 | ≥16 | 2.5 | 10 | 4 | 5 | >10 | ≥4 | 0.3125 | 10 | 32 | 1.25 | >10 | ≥16 | 0.3125 | 10 | 32 |
| R. similis | 2.5 | >10 | ≥8 | 2.5 | >10 | ≥8 | 2.5 | >10 | ≥8 | 0.625 | >10 | ≥32 | >10 | >10 | ≥1 | 2.5 | >10 | ≥8 | 2.5 | >10 | ≥8 | 5 | >10 | ≥4 |

**Table 2. Cytotoxicity values, minimal inhibitory concentration (MIC), and Selectivity Index (SI) of selected compounds with activity against the chromoblastomycosis agents.**

| Species | Trifloxystrobin (MMV688754) | | | MMV021013 | | | Azoxystrobin (MMV021057) | | | MMV688179 | | | Difenoconazole (MMV688943) | | | Iodoquinol (MMV002817) | | | Bitertanol (MMV688942) | | |
|---|---|---|---|---|---|---|---|---|---|---|---|---|---|---|---|---|---|---|---|---|---|
| | CC$_{50}$[a] | MIC | SI | CC$_{50}$[b] | MIC | SI | CC$_{50}$[a] | MIC | SI | CC$_{50}$[c] | MIC | SI | CC$_{50}$[a] | MIC | SI | CC$_{20}$[b] | MIC | SI | CC$_{50}$[a] | MIC | SI |
| *C. carrionii* | >64 | 2.5 | 25.6 | >50 | 5 | 10 | >28 | 2.5 | 11.2 | 1 | 5 | 0.2 | 30.9 | 0.625 | 49.4 | 2.5 | 1.25 | 2 | >64 | 1.25 | 51.2 |
| *P. verrucosa* | >64 | 2.5 | 25.6 | >50 | 5 | 10 | >28 | 0.3125 | 89.6 | 1 | 5 | 0.2 | 30.9 | 2.5 | 12.3 | 2.5 | 1.25 | 2 | >64 | 5 | 12.8 |
| *E. dermatitidis* | >64 | 10 | 6.4 | >50 | 5 | 10 | >28 | 10 | 2.8 | 1 | >10 | 0.1 | 30.9 | 2.5 | 12.3 | 2.5 | 1.25 | 2 | >64 | 10 | 6.4 |
| *E. jeanselmei* | >64 | 0.625 | 102.4 | >50 | 5 | 10 | >28 | 0.3125 | 89.6 | 1 | >10 | 0.1 | 30.9 | >10 | 3 | 2.5 | 1.25 | 2 | >64 | 5 | 12.8 |
| *F. pedrosoi* | >64 | 2.5 | 25.6 | >50 | 2.5 | 20 | >28 | 1.25 | 22.4 | 1 | 5 | 0.2 | 30.9 | 1.25 | 24.7 | 2.5 | 1.25 | 2 | >64 | 0.625 | 102.4 |
| *F. monophora* | >64 | 2.5 | 25.6 | >50 | 2.5 | 20 | >28 | 2.5 | 11.2 | 1 | 5 | 0.2 | 30.9 | 2.5 | 12.3 | 2.5 | 1.25 | 2 | >64 | 2.5 | 25.6 |
| *F. nubica* | >64 | 0.625 | 102.4 | >50 | 1.25 | 40 | >28 | 2.5 | 11.2 | 1 | 5 | 0.2 | 30.9 | 0.3125 | 98.8 | 2.5 | 1.25 | 2 | >64 | 0.3125 | 204.8 |
| *R. similis* | >64 | 2.5 | 25.6 | >50 | 2.5 | 20 | >28 | 0.625 | 44.8 | 1 | >10 | 0.1 | 30.9 | 2.5 | 12.3 | 2.5 | 2.5 | 1 | >64 | 5 | 12.8 |

Auranofin: not determined (reference compound); CC$_{50}$ (concentration that killed 50% of the cells) and CC$_{20}$ (concentration that killed 20% of the cells) values were provided by MMV; Data was obtained with the following cell lines: [a] MRC5 cells; [b] HepG2 cells; [c] PMM cells. SI: selectivity index = CC$_{50}$ (μM) / MIC (μM). The higher is the ratio obtained, the more selective is the compound against the fungal pathogen.

are detailed in Tables 3 and 4, respectively. The auranofin and itraconazole combination produced the single synergistic interaction (FICI < 0.5), against the *C. carrionii* strain. The combinations of iodoquinol, bitertanol, and MMV021013 with itraconazole produced indifferent interactions (FICI between 0.5 and 4) for all isolates tested. All compounds in combination with terbinafine produced indifferent interactions.

## Discussion

CBM is difficult to treat because there is no standardized drug of choice. In addition, relapses are frequent for this mycosis [13]. The taxonomic diversity of the CBM agents adds another

**Table 3. Interaction of selected compounds with itraconazole.**

| Species | MIC (μg/ml) | | | FICI AUR/ITZ | MIC (μg/ml) | | | FICI IOD/ITZ | MIC (μg/ml) | | | FICI BIT/ITZ | MIC (μg/ml) | | | FICI MMV021013/ITZ |
|---|---|---|---|---|---|---|---|---|---|---|---|---|---|---|---|---|
| | AUR | ITZ | AUR/ITZ | | IOD | ITZ | IOD/ITZ | | BIT | ITZ | BIT/ITZ | | MMV021013 | ITZ | MMV021013/ITZ | |
| *C. carrionii* | 1.5 | 0.5 | 0.375/0.06 | **0.37*** | 0.25 | 0.5 | 0.007/0.5 | **1.03** | 0.375 | 0.12 | 0.005/0.06 | **0.51** | 2.5 | 0.5 | 1.25/0.12 | **0.74** |
| *P. verrucosa* | 1.5 | 0.25 | 0.375/0.12 | **0.73** | 0.25 | 0.25 | 0.25/0.06 | **1.24** | 3.0 | 0.25 | 1.5/0.12 | **0.98** | 2.5 | 0.25 | 1.25/0.06 | **0.74** |
| *E. dermatitidis* | 1.5 | 0.25 | 0.75/0.06 | **0.74** | 0.5 | 0.25 | 0.25/0.06 | **0.8** | 1.5 | 0.12 | 0.75/0.06 | **1** | 2.5 | 0.5 | 0.003/1.0 | **2** |
| *E. jeanselmei* | 0.75 | 0.12 | 0.09/0.06 | **0.62** | 0.5 | 0.06 | 0.0078/0.06 | **1** | 1.5 | 0.12 | 0.75/0.06 | **1** | 1.25 | 0.06 | 0.003/0.06 | **1** |
| *F. pedrosoi* | 0.75 | 0.12 | 0.01/0.12 | **1** | 0.25 | 0.12 | 0.25/0.015 | **1.12** | 0.375 | 0.12 | 0.01/0.06 | **0.52** | 1.25 | 0.12 | 0.625/0.06 | **1** |
| *F. monophora* | 0.75 | 0.06 | 0.01/0.06 | **1** | 0.25 | 0.12 | 0.125/0.06 | **1** | 0.1875 | 0.12 | 0.187/0.06 | **1.5** | 2.5 | 0.06 | 0.003/0.06 | **1** |
| *F. nubica* | 0.375 | 0.12 | 0.01/0.12 | **1** | 0.25 | 0.12 | 0.125/0.06 | **1** | 0.1875 | 0.12 | 0.09/0.06 | **1** | 0.3125 | 0.12 | 0.15/0.06 | **0.98** |
| *R. similis* | 0.75 | 0.25 | 0.75/0.06 | **1.24** | 0.5 | 0.12 | 0.25/0.06 | **1** | 3.0 | 0.25 | 0.375/0.25 | **1.12** | 2.5 | 0.25 | 0.15/0.12 | **0.54** |

AUR: auranofin; ITZ: itraconazole; IOD: iodoquinol; BIT: bitertanol; FICI: fractional inhibitory concentration index.

* indicates a synergistic combination.

**Table 4. Interaction of selected compounds with terbinafine.**

| Species | MIC (µg/ml) | | | FICI AUR/TRB | MIC (µg/ml) | | | FICI IOD/TRB | MIC (µg/ml) | | | FICIBIT/TRB | MIC (µg/ml) | | | FICI MMV021013/TRB |
|---|---|---|---|---|---|---|---|---|---|---|---|---|---|---|---|---|
| | AUR | TRB | AUR/TRB | | IOD | TRB | IOD/TRB | | BIT | TRB | BIT/TRB | | MMV021013 | TRB | MMV021013/TRB | |
| *C. carrionii* | 0.75 | 0.25 | 0.75/0.015 | 1.06 | 0.50 | 0.50 | 0.125/0.25 | 0.75 | 0.75 | 0.12 | 0.375/0.12 | 0.98 | 2.5 | 0.25 | 2.5/0.12 | 1.48 |
| *P. verrucosa* | 0.75 | 0.06 | 0.75/0.015 | 1.25 | 0.25 | 0.12 | 0.125/0.06 | 1 | 1.5 | 0.12 | 1.5/0.06 | 1.5 | 2.5 | 0.12 | 1.25/0.015 | 0.62 |
| *E. dermatitidis* | 0.75 | 0.25 | 0.75/0.015 | 1.06 | 0.5 | 0.25 | 0.25/0.12 | 0.98 | 1.5 | 0.25 | 1.5/0.12 | 1.48 | 5.0 | 0.25 | 0.625/0.25 | 1.12 |
| *E. jeanselmei* | 1.5 | 0.5 | 0.75/0.06 | 0.62 | 0.50 | 0.50 | 0.25/0.03 | 0.56 | 1.5 | 0.25 | 0.75/0.12 | 0.98 | 2.5 | 0.50 | 1.25/0.25 | 1 |
| *F. pedrosoi* | 0.75 | 0.06 | 0.375/0.03 | 1 | 0.50 | 0.12 | 0.25/0.03 | 0.75 | 0.375 | 0.12 | 0.187/0.06 | 1 | 2.5 | 0.06 | 0.003/0.06 | 1 |
| *F. monophora* | 1.5 | 0.12 | 0.375/0.12 | 1 | 0.50 | 0.12 | 0.25/0.03 | 0.75 | 0.1875 | 0.25 | 0.09/0.06 | 0.98 | 2.5 | 0.12 | 2.5/0.015 | 1.12 |
| *F. nubica* | 0.375 | 0.12 | 0.005/0.12 | 1 | 0.25 | 0.25 | 0.25/0.015 | 1 | 0.1875 | 0.12 | 0.09/0.015 | 0.62 | 1.25 | 0.12 | 0.625/0.015 | 0.62 |
| *R. similis* | 0.75 | 1.0 | 0.75/0.50 | 1.50 | 0.5 | 1.0 | 0.5/0.5 | 1.50 | 3.0 | 1.0 | 1.5/0.5 | 1 | 2.5 | 1.0 | 2.5/0.50 | 1.50 |

AUR: auranofin; TRB: terbinafine; IOD: iodoquinol; BIT: bitertanol; FICI: fractional inhibitory concentration index.

obstacle to the development of new drugs against this disease, because most patients are diagnosed exclusively on the basis of the presence of muriform cells in infected tissues [2]. Therefore, general therapeutic decisions are made regardless of the etiology of CBM, which may be deleterious for the patient. In this study, for instance, *E. dermatitidis* often presented MIC values higher than other species for the studied drugs. In this context, we recommend that repurposing studies for CBM use a large number of etiologic agents of this disease in the antifungal activity experiments.

The compound collection used in this study, the Pathogen Box® library, was previously demonstrated to contain molecules with antifungal activity. Mayer and Kronstad identified five compounds (tolfenpyrad, difenoconazole, bitertanol, posaconazole, and MMV688271) that exhibited antifungal activity against *Cryptococcus neoformans* and *Candida albicans* [18]. In our study, antifungal activities were identified for three of these compounds (difenoconazole, posaconazole, and bitertanol). It is noteworthy that tolfenpyrad inhibited *F. pedrosoi* growth by 49%. This compound is an agricultural insecticide and with well-known poisoning effects [30]. MMV688271, which completely inhibited the growth of *C. neoformans* and *C. albicans* [18], showed only 36% inhibition against *F. pedrosoi* in our screening assay. Vila and Lopez-Ribot evaluated three of the compounds that we investigated (MMV687807, MMV687273, and MMV688768) against *C. albicans* biofilms [17]. In the present study, these compounds showed 13% (MMV687807), 37% (MMV687273) and 24% (MMV688768) inhibition of the *F. pedrosoi* growth. Wall et al identified three compounds with anti-*C. auris* activity in the Pathogen Box® at a 5 µM concentration, including pentamidine, MMV659010, and iodoquinol. In this group, iodoquinol and MMV659010 showed %IG of 86 and 62, respectively, against plant cells. Pentamidine showed 61% inhibition in biofilms of *C. auris* [31]. In the present study, these compounds presented 34% (MMV659010), 49% (pentamidine), and 99% (iodoquinol) inhibition of the *F. pedrosoi* growth. These results clearly demonstrate that major fungal pathogens respond differently to the Pathogen Box® compounds (S1 Table).

The MMV Pathogen Box® contains two antifungal drugs as reference compounds. Inhibition of *F. pedrosoi* growth by posaconazole (99%) was observed, but amphotericin B inhibited

less fungal growth (29%). This data can be explained by the high MIC values of amphotericin B (4–16 μg/ml) against this species [14].

In our study, three compounds were selected as promising anti-CBM agents: auranofin, iodoquinol, and MMV021013. Auranofin has been in clinical use as the gold standard for the oral treatment of rheumatoid arthritis since the 1980s [32,33]. The oral bioavailability of auranofin and its reduced side effects offer significant advantages over traditional injectable drugs [32,33]. Several studies were conducted to identify alternative therapeutic applications for auranofin, particularly in the area of infectious diseases. For example, antifungal activity against *C. albicans* [34] and *C. neoformans* [34–36] has been described. In this study, low MIC values for this drug against eight different CBM agents were observed. Furthermore, the synergism found in the combination of auranofin with itraconazole against *C. carrionii*, the main CBM agent in arid and semi-arid areas [37], as well as its fungicidal activity against *E. dermatitidis* support the use of auranofin as a therapeutic option for CBM. These findings justify future *in vivo* testing and clinical studies to evaluate auranofin as an antifungal drug.

Iodoquinol (diiodohydroxyquinoline) is a luminal amebicide. Its use is recommended after treatment with a tissue amebicide, such as metronidazole, to eradicate surviving parasites in the colon [38]. This drug has low absorption through the gastrointestinal tract, so only the topical formulation would be applicable against CBM. Burnett and Mitchell demonstrated that the topical formulation of 1% iodoquinol in combination with 2% hydrocortisone acetate had antifungal activity against *C. albicans*, dermatophytes, and *Malassezia furfur* [39]. The study by Wall et al. demonstrated the *in vitro* efficacy of iodoquinol against several *Candida* species under planktonic conditions. As expected from the initial results with *C. auris*, iodoquinol was virtually ineffective against preformed biofilms of all species tested. However this compound displayed potent activity against planktonic growth of all strains tested, with $IG_{50}$ (50% inhibition growth) values generally lower than 1 μg/ml [31]. In the present study, lower MIC values of iodoquinol (100% inhibition) were observed against all studied CBM agents, supporting its use against this group of fungi.

MMV021013 is a 2-pyridyl-4-aminopyrimidine. Antifungal activity has been described for some aminopyrimidines, including4-(2-Fluorophenyl)-6-trifluoromethyl-2-aminopyrimidine, active against *Botrytis cinera* [40], and 4-(1H-benzo[d]imidazol-2-yl)-6-(p-tolyl)pyrimidin-2-amine, active against *Aspergillus flavus*, *Candida utilis*, and *Saccharomyces cerevisiae* [41]. The Pathogen Box® classifies this compound as a tuberculosis targeted drug. Also, it was found to be active against *Leishmania donovani* and *Trypanosoma cruzi* [19]. To the best of our knowledge, no antifungal activity has been described for MMV021013. Some studies have identified molecules with *in vitro* activity against mycotic and tuberculosis agents [42,43], as well as against protozoa and fungi [44]. These observations suggest that these pathogens may share common antimicrobial targets that may propel drug discovery against a broad range of infectious diseases.

In summary, auranofin was identified in our study as a promising compound to be tested as a CBM therapeutic agent, on the basis of its low MIC values against CBM agents, synergism with itraconazole against a *C. carrionii* strain, and fungicidal action against *P. verrucosa* and *E. dermatitidis*. Iodoquinol has also proved promising for future studies involving CBM agents due to its low MIC values, fungicidal activity against *P. verrucosa*, and approved topical use in some countries. The 2-pyridyl-4-aminopyrimidine MMV021013 showed good selectivity and a combined activity against mycobacteria, *Leishmania*, *Trypanosoma*, and CBM agents. These data suggest the need for further *in vitro* and *in vivo* studies with these drugs aiming to develop new therapeutic tools against the CBM agents.

## Supporting information

**S1 Table. Screening of 400 compounds present in the MMV Pathogen Box® collection against a *Fonsecaea pedrosoi* strain (CFP00791).** Results are presented as the percent inhibition of growth (%IG) of the fungal strain.
(PDF)

## Acknowledgments

We are grateful for support from the Coordination for the Improvement of Higher Education Personnel (CAPES). We thank Medicines for Malaria Venture (MMV, Switzerland) for support, design and supply of the Pathogen Box®. We thank for Ingrid Ludmilla Rodrigues da Cruz, Iara Bastos de Andrade, Mônica dos Santos Elias, and Jonas Campos Pereira that contributed with technical assistance in this study.

## Author Contributions

**Conceptualization:** Rowena Alves Coelho, Rodrigo Almeida-Paes.

**Data curation:** Rowena Alves Coelho, Maria Helena Galdino Figueiredo-Carvalho.

**Formal analysis:** Luna Sobrino Joffe, Fábio Brito-Santos, Marcio L. Rodrigues, Rodrigo Almeida-Paes.

**Funding acquisition:** Rodrigo Almeida-Paes.

**Methodology:** Rowena Alves Coelho, Luna Sobrino Joffe, Gabriela Machado Alves, Maria Helena Galdino Figueiredo-Carvalho.

**Project administration:** Marcio L. Rodrigues, Rodrigo Almeida-Paes.

**Resources:** Marcio L. Rodrigues, Rodrigo Almeida-Paes.

**Supervision:** Rodrigo Almeida-Paes.

**Validation:** Marcio L. Rodrigues, Rodrigo Almeida-Paes.

**Visualization:** Ana Claudia Fernandes Amaral.

**Writing – original draft:** Rowena Alves Coelho, Rodrigo Almeida-Paes.

**Writing – review & editing:** Maria Helena Galdino Figueiredo-Carvalho, Fábio Brito-Santos, Ana Claudia Fernandes Amaral, Marcio L. Rodrigues, Rodrigo Almeida-Paes.

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
