## [Decision Letter · Decision Letter 0]

3 Mar 2020

PONE-D-20-03641

A screening of the MMV Pathogen Box® reveals new potential antifungal drugs against the etiologic agents of chromoblastomycosis

PLOS ONE

Dear Mrs Coelho,

Thank you for submitting your manuscript to PLOS ONE. After careful consideration, we feel that it has merit but does not fully meet PLOS ONE’s publication criteria as it currently stands. Therefore, we invite you to submit a revised version of the manuscript that addresses the points raised during the review process.

REQUIRED: Both reviewers note confusion related to association between fungistatic vs. fungicidal determinations and recommended clarification of the text related to these studies. Please clarify this text to make sure the studies are easy to understand for a lay reader.

REQUIRED: Respond to all reviewer comments in a point-by-point response.

We would appreciate receiving your revised manuscript by Apr 17 2020 11:59PM. To enhance the reproducibility of your results, we recommend that if applicable you deposit your laboratory protocols in protocols.io, where a protocol can be assigned its own identifier (DOI) such that it can be cited independently in the future. For instructions see: http://journals.plos.org/plosone/s/submission-guidelines#loc-laboratory-protocols

We look forward to receiving your revised manuscript.

Kind regards,

Kirsten Nielsen, Ph.D

Academic Editor

PLOS ONE

Journal Requirements:

Reviewers' comments:

Reviewer's Responses to Questions

**Comments to the Author**

1. Is the manuscript technically sound, and do the data support the conclusions?

Reviewer #1: Partly

Reviewer #2: Yes

2. Has the statistical analysis been performed appropriately and rigorously? 

Reviewer #1: N/A

Reviewer #2: N/A

3. Have the authors made all data underlying the findings in their manuscript fully available?

Reviewer #1: Yes

Reviewer #2: Yes

4. Is the manuscript presented in an intelligible fashion and written in standard English?

Reviewer #1: Yes

Reviewer #2: Yes

5. Review Comments to the Author

Reviewer #1: In the manuscript “A screening of the MMV Pathogen Box® reveals new potential antifungal drugs against the etiologic agents of chromoblastomycosis” the authors use the MMV Pathogen Box to identify novel treatments for CBM. An initial screening of 400 compounds against F. pedrosi identified 8 compounds that exhibited antifungal activity. The efficacy of these 8 compounds were then tested against other CBM agents. Additionally, the 8 identified compounds were tested for synergy with itraconazole and terbinafine (two compounds that are commonly used to treat CBM), which ultimately led to the identification of three promising compounds. Throughout the paper, the authors provide good explanations of the methodology, the inclusion of relevant formulas was especially interesting. The analysis provided in this paper would be beneficial for further inquiry into novel CBM therapies. Overall, the paper is well written, and the analyses are explained and presented well.

Major points:

L116: 1% DMSO seems like a high percentage of DMSO, and it could be inhibitory alone to the cells -- was a DMSO control included? If so, this should be included in the text, if not the experiments should be repeated to verify 1% DMSO is not inhibitory or justification for why it was not should be explained.

Table 1: Please provide some clarification on why the MFC value is lower than the MIC value for auranofin against P. verrucosa and for bitertanol against P. verrucosa. And why they are still considered fungistatic? Denoting which drugs were fungicidal would also be helpful in this table, so they are readily identifiable.

L144 -145: “Compounds that presented a MFC value lower than two times the MIC were considered fungicidal.” I am confused by this and need more clarification, especially how this relates to the fungicide determination presented in Table 1. In the results (lines 195-196) the authors state that auranofin is fungicidal against E. dermatidis and iodoquinol is fungicidal against P. verrucosa, but the MFC and MIC values are the same. Perhaps explaining fungicidal determination in terms of MFC to MIC ratio would make it easier for the reader to understand and would be less open to interpretation.

Minor points:

Why was F. pedrosi chosen as the strain to test initially? More background on CBM would be interesting to include in the introduction.

L44-45: Include in the abstract why MMV021013 was thought to be promising. It is included for the other two, but not for the MMV021013.

L65: delete “for”, so it reads “time to seek medical help”

L81: italicize in vitro

L148: Define CC50 in text.

L266 & 267: Keep spelling of amebicide consistent.

Figure 1: I liked that you the compounds were grouped by original application, but it would be nice if the 8 compounds that were studied further were labelled. The markers could also be made smaller just to make it easier to see all the markers.

Figure 2: I am not sure this figure is necessary – perhaps it could be included as a supplementary figure.

Table 2, L478: Unsure what “* Cytotoxicity dada only CC20 Hep G2 cells” means. – Is “dada” supposed to be data?

Reviewer #2: The aims of this study was to identify the in vitro antifungal activity of compounds from the Pathogen Box against Chromoblastomycosis agents. Eight strains from the genera Fonsecaea, Cladophialophora, Rhinocladiella, Phialophora, Exophiala, were tested. Initially 400 compounds were tested against Fonsecaea pedrosoi. Eight compounds were found to have a high inhibition activity against F. pedrosoi were then tested for their MIC and MFC against the 8 CBM agents. After excluding compounds with higher MIC, lower SI and toxicity, 4 final compounds were selected to be tested for a synergistic activity with standards antifungal drugs used to treat CBM. In combination with itraconazole, only one compound (Auranofin) showed synergy against C. carrionii but indifference for other CBM agents. All compounds in combination with terbinafine produced indifferent interactions.

In conclusion, this study identified 3 promising compounds (Auranofin, Iodoquinol and MMV021013) with an anti-CBM activity. This is a well written manuscript and the conclusions are supported by the findings. Indicated below are few question and minor errors that need addressing:

1. Is Auranofin fungistatic against P. verrucosa? Lines 195-196 say that “Auranofin (MMV688978) was fungicidal against the E. dermatitidis strain and 196 fungistatic against the other species”. The definition of a fungicidal drug on line 144 indicate that “Compounds that presented a MFC value lower than two times the MIC were considered fungicidal”. For P. verrucosa MIC=1.25 AND mfc=0.625. Unless I am mistaken I think this MFC value indicate that Auranofin is fungicidal against P. verrucosa too.

2. I will suggest to add what “CC50/CC20” means below Table 2.

3. Minor corrections:

a. Line 81, “in vitro” should be in italic

b. Line 182, add a period before “The chemical structure…”.

c. Line 478, correct “cytotoxicity dada”

6. PLOS authors have the option to publish the peer review history of their article (what does this mean?). If published, this will include your full peer review and any attached files.

Reviewer #1: Yes: Sophie Altamirano

Reviewer #2: No

---

## [Author Response · Author response to Decision Letter 0]

8 Apr 2020

Major points:

Reviewer #1: L116: 1% DMSO seems like a high percentage of DMSO, and it could be inhibitory alone to the cells -- was a DMSO control included? If so, this should be included in the text, if not the experiments should be repeated to verify 1% DMSO is not inhibitory or justification for why it was not should be explained.

Answer: We included a DMSO control on the test board, since we follow the EUCAST protocol that recommends a concentration of 1% DMSO during preparation of the microplates for susceptibility testing. After addition of the fungal inoculum, this concentration drops to 0.5%, which is the final DMSO concentration in the antifungal susceptibility testing. This information was clarified in the new version of the text (lines 122-124 and lines 127-131). 

Reviewer #1: Table 1: Please provide some clarification on why the MFC value is lower than the MIC value for auranofin against P. verrucosa and for bitertanol against P. verrucosa. And why they are still considered fungistatic? Denoting which drugs were fungicidal would also be helpful in this table, so they are readily identifiable.

Answer: Thank you for this observation. The MFC lower than the MIC values are really intriguing, therefore we prepared two additional replicates using these compounds against the P. verrucosa isolate. We found MIC values lower or equal to the MFC values. We believe that this one-fold dilution variation on our MFC values were due to intrinsic aspects of the methodology, which has several pipetting and diluting steps that can induce manual errors in the measurements. In fact, several publications only consider two-fold variations on MIC/MFC values as significant. Again, thank you for this observation.

Reviewer #1: L144 -145: “Compounds that presented a MFC value lower than two times the MIC were considered fungicidal.” I am confused by this and need more clarification, especially how this relates to the fungicide determination presented in Table 1. In the results (lines 195-196) the authors state that auranofin is fungicidal against E. dermatidis and iodoquinol is fungicidal against P. verrucosa, but the MFC and MIC values are the same. Perhaps explaining fungicidal determination in terms of MFC to MIC ratio would make it easier for the reader to understand and would be less open to interpretation.

Answer: Thank you for this suggestion. We have changed the definition (lines 152-154) accordingly and also the MFC/MIC ratio was included in table 1 to facilitate the readers’ understanding. Results of fungicidal drugs were also updated in this new version of the manuscript (lines 224-228). Moreover, as mentioned before, we repeated the test twice to confirm the results after receiving the reviewer’s comments.

Reviewer #2: Is Auranofin fungistatic against P. verrucosa? Lines 195-196 say that “Auranofin (MMV688978) was fungicidal against the E. dermatitidis strain and 196 fungistatic against the other species”. The definition of a fungicidal drug on line 144 indicate that “Compounds that presented a MFC value lower than two times the MIC were considered fungicidal”. For P. verrucosa MIC=1.25 AND MFC=0.625. Unless I am mistaken I think this MFC value indicate that Auranofin is fungicidal against P. verrucosa too.

Answer: We followed the helpful suggestion of reviewer #1 to address this problem. In the new version of our manuscript we corrected the results related to fungicidal effects of the evaluated drugs against P. verrucosa (lines 224-228).

Reviewer #2: I will suggest to add what “CC50/CC20” means below Table 2.

Answer: Thank you for this suggestion. These definitions were added in table 2 footnotes as requested.

Minor points:

Reviewer #1: Why was F. pedrosoi chosen as the strain to test initially? More background on CBM would be interesting to include in the introduction.

Answer: We agree with the reviewer. A description of the species epidemiology on the major endemic areas was included in the first paragraph of introduction (Lines 59-65). In addition, the explanation for using this species in all assays, including the screening, was included in the “fungal strains and culture conditions” section of Materials and methods (Lines 95-96).

Reviewer #1: L44-45: Include in the abstract why MMV021013 was thought to be promising. It is included for the other two, but not for the MMV021013.

Answer #1: We added a sentence about this subject in lines 44-45.

Reviewer #1: L65: delete “for”, so it reads “time to seek medical help”

Answer #1: This was corrected as requested.

Reviewer #1: L81: italicize in vitro

Answer #1: This was corrected as requested.

Reviewer #1: L148: Define CC50 in text.

Answer #1: This definition was included in the new version of the manuscript (lines 157-158)

Reviewer #1: L266 & 267: Keep spelling of amebicide consistent.

Answer: This was corrected as requested.

Reviewer #1: Figure 1: I liked that you the compounds were grouped by original application, but it would be nice if the 8 compounds that were studied further were labelled. The markers could also be made smaller just to make it easier to see all the markers.

Answer: A revised Fig1 file was included in the new version of our manuscript. In this new version of Fig 1, we used colors to differentiate the compounds that were further studied. Also, we used smaller markers and increased the sizes of both x and y axis, to facilitate the visualization of the points. However, there are some classes of compounds (e.g. anti-malaria) with a lot of data points, so it is still difficult to differentiate most of them. But, in our opinion, in this new version of the figure, the compounds with anti-CBM activity are clearly differentiated. We sincerely wish that this new Fig1 file will please the reviewers.

Reviewer #1: Figure 2: I am not sure this figure is necessary – perhaps it could be included as a supplementary figure.

Answer: We preferred to maintain this figure within the manuscript, as usually practiced in similar studies. Moreover, reviewer #2 made a minor comment on our text about the chemical structure of the compounds, so we suppose that the other reviewer also thinks that this figure fits within the main text of our manuscript.

Reviewer #1: Table 2, L478: Unsure what “* Cytotoxicity dada only CC20 Hep G2 cells” means. – Is “dada” supposed to be data?

Answer: This legend was corrected (Line 291). Thanks for this observation.

Reviewer #2: Line 81, “in vitro” should be in italic

Answer: This was corrected as requested.

Reviewer #2: Line 182, add a period before “The chemical structure…”.

Answer: This was corrected as requested.

Reviewer #2: Line 478, correct “cytotoxicity dada”

Answer: This was corrected as requested.

We are hopeful that the revised manuscript will be suitable for publication in PLoS One.

Sincerely,

Rowena Alves Coelho.

---

## [Decision Letter · Decision Letter 1]

23 Apr 2020

PONE-D-20-03641R1

A screening of the MMV Pathogen Box® reveals new potential antifungal drugs against the etiologic agents of chromoblastomycosis

PLOS ONE

Dear Mrs Coelho,

Thank you for submitting your manuscript to PLOS ONE. After careful consideration, we feel that it has merit but does not fully meet PLOS ONE’s publication criteria as it currently stands. Therefore, we invite you to submit a revised version of the manuscript that addresses the points raised during the review process.

The reviewers noted a few very minor edits that were not incorporated into manuscript so it has been returned to correct these errors prior to final acceptance. A detailed response is not required, just acknowledgement that the corrections have been made to the submitted manuscript.

We would appreciate receiving your revised manuscript by Jun 07 2020 11:59PM. To enhance the reproducibility of your results, we recommend that if applicable you deposit your laboratory protocols in protocols.io, where a protocol can be assigned its own identifier (DOI) such that it can be cited independently in the future. For instructions see: http://journals.plos.org/plosone/s/submission-guidelines#loc-laboratory-protocols

We look forward to receiving your revised manuscript.

Kind regards,

Kirsten Nielsen, Ph.D

Academic Editor

PLOS ONE

Reviewers' comments:

Reviewer's Responses to Questions

**Comments to the Author**

1. If the authors have adequately addressed your comments raised in a previous round of review and you feel that this manuscript is now acceptable for publication, you may indicate that here to bypass the “Comments to the Author” section, enter your conflict of interest statement in the “Confidential to Editor” section, and submit your "Accept" recommendation.

Reviewer #1: All comments have been addressed

Reviewer #2: (No Response)

2. Is the manuscript technically sound, and do the data support the conclusions?

Reviewer #1: Yes

Reviewer #2: Yes

3. Has the statistical analysis been performed appropriately and rigorously? 

Reviewer #1: N/A

Reviewer #2: N/A

4. Have the authors made all data underlying the findings in their manuscript fully available?

Reviewer #1: Yes

Reviewer #2: Yes

5. Is the manuscript presented in an intelligible fashion and written in standard English?

Reviewer #1: Yes

Reviewer #2: Yes

6. Review Comments to the Author

Reviewer #1: All of my comments were addressed, and I think the authors made all of the necessary revisions.

Minor

L154: Change fungicide to fungicidal

Reviewer #2: The authors has addressed most of comments on the previous version of the manuscript. However, modifications added on lines 225-226 were not added in the final summary (lines 381-382). I will suggest adding in the final summary that auranofin is also fungicidal against P. verrucosa.

7. PLOS authors have the option to publish the peer review history of their article (what does this mean?). If published, this will include your full peer review and any attached files.

Reviewer #1: No

Reviewer #2: No

---

## [Author Response · Author response to Decision Letter 1]

24 Apr 2020

After the last review the changes suggested by reviewers # 1 and # 2 have been corrected. Thanks for this observation.

Reviewer #1: All of my comments were addressed, and I think the authors made all of the necessary revisions.

Minor

L154: Change fungicide to fungicidal 

Answer: This was corrected as requested.

Reviewer #2: The authors has addressed most of comments on the previous version of the manuscript. However, modifications added on lines 225-226 were not added in the final summary (lines 381-382). I will suggest adding in the final summary that auranofin is also fungicidal against P. verrucosa.

Answer: This was corrected as requested.

---

## [Editor Report · Decision Letter 2]

28 Apr 2020

A screening of the MMV Pathogen Box® reveals new potential antifungal drugs against the etiologic agents of chromoblastomycosis

PONE-D-20-03641R2

Dear Dr. Coelho,

We are pleased to inform you that your manuscript has been judged scientifically suitable for publication and will be formally accepted for publication once it complies with all outstanding technical requirements.

With kind regards,

Kirsten Nielsen, Ph.D

Academic Editor

PLOS ONE
---

## [Editor Report · Acceptance letter]

30 Apr 2020

PONE-D-20-03641R2 

A screening of the MMV Pathogen Box® reveals new potential antifungal drugs against the etiologic agents of chromoblastomycosis 

Dear Dr. Coelho:

I am pleased to inform you that your manuscript has been deemed suitable for publication in PLOS ONE. Congratulations! Your manuscript is now with our production department. 

With kind regards,

on behalf of

Dr. Kirsten Nielsen 

Academic Editor

PLOS ONE